# Predicting the Properties of Industrially Produced Oat Flours by the Characteristics of Native Oat Grains or Non-Heat-Treated Groats

**DOI:** 10.3390/foods10071552

**Published:** 2021-07-05

**Authors:** Iina Jokinen, Juha-Matti Pihlava, Anna Puganen, Tuula Sontag-Strohm, Kaisa M. Linderborg, Ulla Holopainen-Mantila, Veli Hietaniemi, Emilia Nordlund

**Affiliations:** 1VTT Technical Research Centre of Finland, Ltd., P.O. Box 1000, FI-02044 Espoo, Finland; ulla.holopainen@vtt.fi (U.H.-M.); emilia.nordlund@vtt.fi (E.N.); 2Luke Natural Resources Institute Finland (Luke), FI-31600 Jokioinen, Finland; juha-matti.pihlava@luke.fi (J.-M.P.); veli.hietaniemi@luke.fi (V.H.); 3Food Chemistry and Food Development, Department of Life Technologies, University of Turku, FI-20014 Turku, Finland; anna.puganen@utu.fi (A.P.); kaisa.linderborg@utu.fi (K.M.L.); 4Department of Food and Nutrition, Faculty of Agriculture and Forestry, University of Helsinki, Agnes Sjöberginkatu 2, PL66, FI-00014 Helsinki, Finland; tuula.sontag-strohm@helsinki.fi

**Keywords:** oats, quality, processing, color, composition, NIT

## Abstract

The aim of this study was to determine whether the properties of the native oat grain or non-heat-treated groats (laboratory-scale dehulling) can be used to predict the quality of the industrially produced oat flour produced from heat-treated groats. Quality properties such as the color, hectoliter weight, thousand seed weight and hull content of Finnish native grains (*n* = 30) were determined. Furthermore, the relationship between the properties of the native grains and the chemical composition of the raw oat materials before and after the milling process were studied. A significant relationship (*p* < 0.01) was observed between the thousand seed weight of the native oat groats and the chemical composition of the industrially produced oat flour. Furthermore, the protein content of the native grains measured by NIT correlated with the chemical composition of the oat flours. These results suggest that the properties of oat flour produced on an industrial scale, including heat treatment, could be predicted based on the properties of native oat grains.

## 1. Introduction

The importance of oats is increasing globally due to public health concerns and the need for a shift to a plant-based diet. The worldwide production of oats has been slowly increasing during recent years, with a current production of approximately 23–26 million tons annually [1]. For example, in 2020, 140,000 tons of the total 1.2-million-ton oat production in Finland were processed into foods for domestic and export purposes, which was 6% more than in 2019 [2]. Compared to other cereal grains, oats have a relatively high lipid content, and they are a good source of dietary fibre, proteins, minerals and bioactive compounds [3]. Oats (*Avena sativa* L.) have excellent nutritional value, and they are known for their health benefits in plasma cholesterol lowering and postprandial blood sugar control, which are related to the dietary fibre β-glucan [4].

The chemical composition of oats is known to vary depending on the cultivar, as well as the growing location and conditions [5,6,7,8,9]. The preferences for using oats for food and feed differ in terms of their chemical composition [10]. For example, for food use, a high β-glucan content is often considered desirable, whereas in feed the lower β-glucan content is preferred. The main quality indicators currently used for milling-grade oats are their hectoliter weight, soundness count, color, moisture, foreign material and protein content [11]. Several methods can be used to assess these quality indicators of oat grains. In addition to chemical measurements, methods based on near-infrared spectroscopy, near-infrared reflectance (NIR) and near-infrared transmittance (NIT) can be applied for the faster evaluation of the oats’ quality [12].

In addition to the physical measurements describing the grain size and color, chemical measurements of moisture, protein content and starch properties are used as the most typical indicators of quality for other grains such as wheat [13]. For oats, the content of β-glucan is typically included, as it relates to its health effects in food use. Furthermore, the quantity and composition of lipids is relevant to be included, as the presence and degradation of lipids complicates certain food and feed processes. The color of oat grains or oat products has been indicated to be one of the important factors affecting the consumer acceptance and liking of them [14]. Furthermore, the color has been found to show a significant year-by-crop interaction in oat grains and rolled oats, and a dark color of oat grains is a known quality defect of oats [15,16]. However, the relationship between the oat grain color and the other grain properties to facilitate the approval of oat batches by the industry has not been established before. 

Oat milling has some special features when compared to other cereal grains, as it includes a heat treatment by steaming, also known as kilning, which was developed to inactivate endogenous enzymes that can induce lipid degradation in oat flours [11,17]. Although most of the oats used in the food industry go through the milling process, including kilning, the research regarding the chemical composition often focuses on oat raw materials processed only on a laboratory scale [5,6,8,9,18]. Oats are used in the food industry mostly as flour or rolled oats, and as batches comprising of oats grown in different locations and of different cultivars [11,15]. However, oat-based ingredients are also increasingly used in food products such as bread, dairy or meat alternatives. The increased use in different food sectors poses a need for scientific information on the variation of the chemical composition of oat materials. Such information would help the food industry to optimize their oat raw material selection for the manufacture of different types of oat products. For example, a high β-glucan content is preferred for some applications with health claims based on β-glucan (e.g., full-oat breads), while a high protein content is preferred for some ingredients, such as meat alternatives [10,19].

In this research, the aim was to investigate the ways in which the chemical and physical properties of native oat grains correlate with the properties of non-heat-treated groats (laboratory-scale de-hulling), as well as heat-treated groats and the subsequent oat flour (industrial scale milling process). This was performed by characterizing 30 different oats from Finland, and by performing profound correlation analysis. The ultimate goal was to evaluate whether measurements from the native grains or from the non-heated groats can be used to predict heated oat flour properties after industrial-scale processing, which includes kilning. In particular, we wanted to estimate the potential of rapid analysis methods, such as NIT, to predict the oat flour quality after the milling process. 

## 2. Materials and Methods

### 2.1. Materials

Thirty Finnish oat samples representing 23 oat cultivars were chosen and obtained from Boreal Plant Breeding Ltd., Jokioinen, Finland; Peltosiemen Ltd., Forssa, Finland; Vääksyn Mylly Ltd., Vääksy, Finland; Plantanova Ltd., Ruukki, Finland; Raisio plc, Raisio, Finland; and Lantmännen Agro Ltd., Vantaa, Finland. The samples were chosen based on their availability and on the expectation that they would differ in their composition and physical properties; they were processed as a pure cultivar. Three different crop years—2017, 2018 and 2019—were represented, and some cultivars were included twice from different crop years. The samples were analyzed as native oat grains, as non-heated oat groats dehulled at the laboratory scale, and non-heated flour produced from non-heated groats (Figure 1). In addition, the oats were analyzed as heated oat groats and as oat flour produced by an industrial-scale milling process from the heated oat groats after flaking (Figure 1).

In this article, “grain” denotes the whole intact kernel and “groat” denotes the de-hulled grain. The non-heated groats were produced by de-hulling the native oat grains at the laboratory scale with an oat de-huller (Rivakka, Nipere Ltd., Teuva Finland), and were cleaned using a universal threshing machine (Baumann-Saatzuchtbedarf, Waldenburg, Germany). Approximately 50 g non-heat-treated groats were weighed and ground with an ultra-centrifugal mill (Fritsch Pulverisette 14, Idar-Oberstein, Germany) using a 0.5 mm sieve and 15,000 rpm speed, or with a KT-120 hammer mill (Koneteollisuus Ltd., Klaukkala, Finland) with a 0.5 mm sieve, and were then stored below −18 °C. The heated oat flours were milled at Vääksyn Mylly Ltd. (Asikkala, Finland) to produce the heated groats and the heated oat flake flour. The milling process included drying, de-hulling, kilning, flaking and milling steps (Figure 1).

### 2.2. Chemical and Physical Analyses

#### 2.2.1. Physical Properties

The dry matter content, protein content and hectoliter weight of the native grain samples were determined by NIT in triplicates (Infratec NIT 1241 Grain Analyser, FOSS). The sample amount was approximately 600 g. In addition, the hectoliter weight of the native grains, non-heated groats and heated groats was determined mechanically in triplicates with a Grain tester 1938 (Pfeuffer GmbH, Kitzingen, Germany). The thousand seed weight of the grains and non-heated groat samples were determined with a Contador seed counter in duplicates (Pfeuffer GmbH), and the amount of hulls was calculated as the thousand seed weigh (native grains)–the thousand seed weight (non-heated groats).

The L*, a* and b* values indicate the lightness red–green and blue–yellow intensity of the measured object, respectively, and they can be transformed into chroma and hue values in order to describe the color of the object. The L*, a* and b* parameters of the oat grains, non-heated and heated groats, and the heated oat flours were determined with a Minolta Chroma Meter CR-200 (Minolta Camera Co., Ltd., Osaka, Japan) in reflection mode. The instrument was standardized with a white ceramic plate (Calibration Plate CR-A43). The oat samples were transferred to a transparent petri dish to form a 1-cm layer. The tip of the sensor was positioned against the bottom of the petri dish and five measurements from different points of the bottom were carried out. The result was an average of five replicates of each oat grain, groat and flour sample. The hue and chroma values were generated from the a* and b* values according to the following equations [18]:(1)$$\mathrm{Hue}\;\mathrm{angle}=\arctan\;\left(b/a\right)$$
(2)$$\mathrm{Chroma}={\left(a^{2}+b^{2}\right)}^{1/2}$$

#### 2.2.2. Chemical Properties

The chemical properties were measured from the non-heated and heated oat flours. The moisture content was determined by drying the samples at 105 °C overnight (17 h). The total starch content was analyzed in triplicates according to the method described by Salo and Salmi (1968) [20]. The protein contents were determined with the Kjeldahl method using a Kjeltec TM8400 analyzer according to the Association of Official Analytical Chemists (AOAC) method 2001.11, with an N factor of 6.25. The total dietary fibre content of the oat flours was analyzed in duplicates using AOAC Method 2011.25 using a semi-automated Dietary Fibre Analyser (ANKOMTDF, Makedon, NY, USA). The β-glucan content was determined in duplicates according to AOAC Method 995.16 using a Megazyme assay kit. The total lipid content was determined using a SoxCap TM 2047 in combination with a Soxtec TM 2050 extraction system with a preparatory acid hydrolysis step and diethyl-ether extraction (Foss A/B, Hillerød, Denmark) according to ISO 6492 (Animal feeding stuffs—Determination of fat content.1999). The ash content was measured by burning the samples at 500 °C overnight (17 h). The methods for the proximate composition analysis, and protein, lipid and ash content, respectively, are accredited by the FINAS Finnish Accreditation Service (Helsinki, Finland). Luke laboratories comply with standard EN ISO/IEC 17025:2017.

The relative percentages of saturated fatty acids (SAFA), monounsaturated fatty acids (MUFA) and polyunsaturated fatty acids (PUFA) were analyzed in duplicate from flour made from non-heated groats and heated flake flours. The oat flour samples were extracted by the four-step lipid extraction method: double extraction with MTBE/methanol, extraction with hexane and extraction with methanol, as modified from Multari et al. (2018) [21]. Triheptadecanoin (TAG 17:0) and 1,2-dipentadecanoyl-sn-glycero-3-phosphatidylcholine (PL 15:0) were used as the internal standards (Larodan, Sweden). The extracted lipids were transformed into fatty acid methyl esters (FAMEs) using an acid-catalyzed method [22]. The methylated samples were further analyzed by a Shimadzu Nexis GC-2030 gas chromatograph with an AOC-20i auto-injector and flame ionization detector (Shimadzu Corporation, Japan) equipped with an Agilent JandW GC column DB-23 (60 m × 0.25 mm i.d., liquid film 0.25 μm, Santa Clara, CA, USA) using helium as the carrier gas. The temperature program described by Linderborg et al. (2019) was applied [23]. Supelco 37 Component FAME mix (Supelco, St. Louis, MO, USA) and FAME standard 68D (Nu-Check-Prep, Elysian, MN, USA) were used as the external standards.

### 2.3. Statistical Analysis

All of the results, average deviations and coefficients of variations were calculated using Excel spreadsheet software (Excel 2016, Microsoft, Redmond, WA, USA). The interactions between the different sample parameters were estimated based on the Pearson correlation coefficients. The differences between the physical properties of the grain and groat samples were analyzed with one-way analysis of variance (ANOVA) using Tukey’s honestly significant difference (HSD) (*p* < 0.05) posthoc test. Independent sample *T*-tests (*p* < 0.05) were performed to compare the differences between the chemical composition parameters. The Pearson correlation coefficients, ANOVA and independent sample *T*-tests were carried out with SPSS-software (IBM SPSS Statistics, version 26, IBM, New York, NY, USA). The sample grouping and differentiation were visually observed based on Partial least squares (PLS) regression, which was performed using The Unscrambler (CAMO Software AS, Oslo, Norway) version 10.5.1.

## 3. Results

### 3.1. Properties of the Grain and Groat Samples

The mechanically determined hectoliter weight and thousand seed weight of the oat grains, non-heated oat groats and heated oat groats showed a large variation within the 30 oat samples (Table 1). The average values of the hectoliter and thousand seed values of each are presented in the Appendix A, Table A1. The hull content of the oat grains calculated based on the laboratory-scale de-hulling varied between 14.1 and 67.3%, while the hull content data calculated based on de-hulling at the mill varied from 23.5 to 76.4% (Table 1). On average, the hull content calculated based on the laboratory scale de-hulling was significantly lower (*p* < 0.05) than the hull content calculated based on de-hulling at the mill. The difference between the hull content of the laboratory-scale and mill-scale samples was batch dependent, with the difference varying from 1.4 to 140% (Appendix A, Table A1).

The color values, L* and b*, of the native oat grains were higher, and the a* values were lower compared to the L*, a* and b* values of the non-heated groat samples (Table 2). The milling process affected the color properties of the oat groats, as the L*, a* and b* values of the heated oat groat samples were significantly (*p* < 0.05) different from the color properties of the grains and non-heated groat samples. The heated groat samples had significantly higher L* and b* values, and lower a* values on average. The color properties of the oat flours were significantly (*p* < 0.05) different compared to the oat grain, non-heated groat and heat-treated groat samples, with higher L* and lower a*and b* values, with a lighter and less intense overall color. The average values of the color values of each sample are presented in the Appendix A, Table A2.

### 3.2. Chemical Properties of Non-Heated and Heated Flour Samples

The chemical composition of the non-heated and heated oat flours showed large batch-dependent variation, which is summarized in Table 3 (with the full data shown in Appendix A, Table A3 and Table A4.). The protein content of the native oat grains varied between 9.1 and 15.8% (dm) in the samples (Table 3). The protein content measured using NIT of the oat grains was significantly lower (*p* < 0.05) compared to the protein content determined by the Kjeldahl method of the non-heated and heated oat flours. In general, the largest variation in chemical composition was observed in the protein and starch contents, which varied between 11.4 and 20.5% (dm) and 46.6 and 75.3% (dm) in the non-heated flour samples, and between 10.6 and 19.2% (dm) and 57.8 and 71.9% (dm) in the heated oat flour samples. The milling process did not affect the protein, fat and fatty acid contents of the oats, as the non-heated flours did not show a significant difference compared to the corresponding values of the heated oat flours. On the other hand, the starch content of the heated oat flours was significantly higher (*p* < 0.05), and the β-glucan and ash contents were significantly lower (*p* < 0.05) when compared with the non-heated flour samples.

### 3.3. Statistical Analysis of the Relationship between the Physical and Chemical Properties of the Oat Samples

#### 3.3.1. Correlations

The chemical and physical properties of oat grains, and the chemical composition of non-heated and heated oat flours had several significant interactions with each other (Appendix A, Table A5 and Table A6). Several moderate correlations (*p* < 0.01) were found between the properties of the native grain samples and the chemical composition of the heated oat flours (Table 4). The lightness values (Color value L*) of the native grain samples showed a significant positive correlation with the L* value (*p* < 0.01), MUFA (*p* < 0.01), protein (*p* < 0.05) and lipid (*p* < 0.05) values of the heated oat flours, and a negative correlation with the PUFA (*p* < 0.01) and starch (*p* < 0.05) values of the heated oat flours. The hue values of the native grains correlated positively with the protein (*p* < 0.01) and ash (*p* < 0.01) content, and negatively with the starch (*p* < 0.05) content of heated oat flours, while the b* or chroma values of the native grain showed no correlation with any of the chemical properties of the oat flours (Appendix A, Table A7).

The thousand seed weight of the native grains showed a significant (*p* < 0.01) correlations with several heated oat flour parameters, including positive correlations with starch and PUFA, and a negative correlation with the a* (*p* < 0.01), lipid (*p* < 0.01) and MUFA (*p* < 0.01) values (Table 3). The hectoliter weight of the grains correlated positively with the starch (*p* < 0.05) and negatively with the protein (*p* < 0.05) content and SAFA (*p* < 0.05) proportion of fatty acids of the heated oat flours. The hull content of the oat grains showed inconsistent weak correlations with the color properties of the heated oat flour. The hull content, calculated based on the laboratory scale de-hulling, correlated positively with color value a* (*p* < 0.05) and negatively with the hue value (*p* < 0.05), while the hull content calculated based on industrial-scale de-hulling showed opposite correlations. Similar relationships to that found for oat grains and oat flours were found between the physical properties of the oat grain and the chemical properties of non-heated flour samples as well (Appendix A, Table A5).

#### 3.3.2. Partial Least Squares (PLS) Regression

The partial least squares (PLS) regression of the properties of the native grains and heated oat flour confirmed the interactions between the different oat grain and heated flour parameters which were observed based on the Pearson correlation coefficients. The visual projection of factors 1 and 2 in PLS of the oat grain and heated oat flour parameters is shown in Figure 2, e.g., the thousand seed weight of the native grains grouped with the PUFA and starch values of heated oat flour. Furthermore, the lightness value (L*) of the oat grain grouped with several chemical components of the heated oat flours and the protein content of the oat grain determined by NIT are grouped with the protein content of the heated oat flour.

## 4. Discussion

### 4.1. Variation in the Properties of the Oat Raw Materials

The variation between the chemical composition of the different oat samples (*n* = 30) was expected, as the composition depends on the growth conditions, location, cultivar and variety [15,24,25]. It is important to note that although all of the samples were produced and analysed as pure cultivars, the experimental design used does not reveal linkages to the role of the cultivar, location, crop year or growth conditions; instead, we focused on revealing the factors of industrially produced oat flours in relation to the properties of the native grains. However, the chemical composition results of the oat samples, when analysed as non-heated and heated flours, were in accordance with the previous literature [9,15,26,27,28]. The large variation in the starch and protein contents are supported by the previous literature. The starch content of oat groats is usually around 60% (dm), but can vary from 39% (dm) up to 67% (dm) [9,27,28]. The protein content of oats usually varies between 10 and 20% (dm), but values as high as 25% (dm) have been reported. Peltonen-Sainio et al. [24] noted that the share of protein was higher after de-hulling, with an increase from 12.7% in oat grains to 15.6% in oat groats, which aligns with the current results. 

The calculated hull content of the grains showed large variation, e.g., 14.1–67.3% determined at the laboratory scale, and was significantly different between the mechanically determined value and the value obtained based on the mill data. This difference most likely originated from the two different de-hulling techniques. The previously reported hull content of oats was approximately 25% [16,24]. The hull content of oats and how tightly the hull is attached to the oat grain is cultivar dependent, as well as being related to the grain size and affecting the milling yield of oats [11]. Furthermore, the hull content of oats can vary within a single cultivar depending on the growth season and conditions [24]. In the current study, the raw material selection also included oat batches that have not been considered ideal for oat milling. Furthermore, the milling process was not optimized for the different raw materials, as the aim was to detect the differences in the milling behavior between the pure cultivar batches. These factors most likely explain the substantial variation and high calculated hull content observed for some of the samples. On the other hand, the industrial scale milling process typically includes a size-based classification of the grains prior to de-hulling, which could lead to more optimal de-hulling compared to the laboratory scale.

According to Lapveteläinen et al. [15], oat groat properties are well maintained during processing, e.g., kilning, drying and flaking. The current results agree to some extent with the previous literature, as the chemical composition of the non-heated flours was to some extent different compared to the parallel-heated oat flours. On average, the β-glucan and ash content of the heated oat flours were lower and the starch was higher than in the non-heated flours, meaning that the milling process on the industrial scale reduced the β-glucan and ash content, and increased the starch content. It is not expected that the heat treatment applied during the kilning step of the milling process could have caused the decrease in the β-glucan content, as heat treatment has not been shown to cause the degradation of β-glucan [29]. It could be that in the de-hulling phase of the industrial mill process, the sub-aleurone layer—rich in β-glucan and ash—was removed more efficiently than in the lab-scale dehulling, and thus relatively more of the β-glucan rich part of the grain was removed from the heated groats [30]. As stated above, the calculated hull contents were slightly higher in the industrial scale samples. Taken together, it is important to pay attention to the dehulling process when analyzing the oat composition at the laboratory scale.

### 4.2. Properties of the Native Oat Grains in Relation to the Properties of the Heated Oat Flour

The main target of the present study was to explore the ways in which the properties of native (unprocessed) oat grains and the properties of non-heated groats (laboratory-scale processing) correlate with the properties of heated groats and the subsequent oat flour (industrial-scale processing). The relationships between the un-processed oat grain and processed oat materials have been studied previously, but mainly by comparing the grain parameters to the oat flake parameters [15]. Currently, the use of oats in other ingredient forms than flakes is increasing their popularity, and therefore understanding the grain indicators for flour quality is important. In the present study, several significant interactions between the physical properties of the native oat grains and the chemical properties of heated oat flours were observed. Furthermore, the protein content of the native grains determined with NIT showed a positive correlation (*p* < 0.01) with the protein content of the oat flour. In addition, the protein content measured with NIT showed some correlations with the other properties of the heated oat flours. This was also observed in the partial least square regression projection, in which the protein content of native grains measured by NIT was clustered with the protein, total dietary fibre, β-glucan and ash content of heated oat flour. These observations indicate that the NIT measurement of oat grains could be used to predict the protein content of oat flour after the milling process.

The specific physical properties of the oat grains, i.e., hectoliter and thousand seed weight, determined in the current study are considered to be important quality parameters for oats. Hectoliter weight is a commonly used grade determinant in the oat trade, as it is considered to predict the oat milling quality well [11,16]. However, it has been observed that the relationship between the milling yield and hectoliter weight is cultivar-dependent [24,31]. The mechanically determined hectoliter weight and thousand seed weight of the oat grains were mainly in agreement with the previous literature [15,24], and they showed both positive correlation with the starch content and negative correlation with the protein content of the oat flours. Furthermore, the thousand seed weight of the oat grains showed several significant interactions with the different shares of fatty acids, as well as negative interactions with the total dietary fibre and β-glucan content of the oat flours. The partial least squares regression projection partly supported these observations, as the thousand seed weight was grouped with the starch content and saturated fatty acid content of industrially produced oat flours, and was on the opposite side of the PLS projection compared to the protein content. Nonetheless, the hectoliter weight of the grains did not group clearly with any of the oat flour parameters in the PLS projection. 

These findings suggest that when aiming for oat ingredients with high protein and β-glucan contents, low hectoliter and thousand seed weights could be used as indicators. On the other hand, high hectoliter and thousand seed weights are typically preferred to produce maximal outputs from the cultivation inputs, and they have also been related to improved milling quality. Favoring raw materials with lower hectoliter and thousand seed weight values could lead to decreased feasibility and milling yield. The hectoliter weight determined by NIT was positively connected to the mechanically determined hectoliter weight, but did not show correlation with any of the chemical components of the oat flours. This implies that hectoliter weight determined by NIT does not predict the chemical properties of the oat raw material as well as the mechanical determination.

In the current study, kilning affected all of the color parameters of the oat groats, as heat-treated groats had higher L* and b* values, and lower a* values than the non-heated groats, meaning that the groats were darker, less red and more yellow in color after kilning. Johnson et al. [18] observed that kilning increased the L* value of oat grains while a* and b* were unaffected, meaning that the grains were also darker due to kilning in their study. Therefore, the observations are similar, although in the present study the oats were processed at the industrial scale rather than the laboratory scale, and were de-hulled prior to kilning rather than being kilned with hulls intact. The color properties of the native grains cannot be clearly connected to the color properties of industrially produced oat flour as almost no correlation was observed between the color properties of these two sample types. Only the lightness value of the native grains showed moderate positive correlation with the lightness value of oat flour, but this interaction was not observed in the PLS projection.

The colour values of the oat grains were found to correlate to some extent with the chemical composition of the non-heated flours, as well as with the chemical composition of the oat flour, as the L* and hue values were linked to the starch, protein and ash content. Johnson, Moot and Lindley [18] noted, that there has not been any confirmed correlation between the grain colour and chemical quality of the oats. As previously mentioned, in oat quality grades, a dark grain colour is considered undesirable, as it is an indicator of quality defects during the harvest. The current study suggests that there is a clear correlation between the grain lightness (colour value L*) and certain chemical compounds of oat flours, thus confirming grain lightness to be an important quality indicator. It has been identified that low a* and high hue values of oat groats are related to higher industrial consumer acceptance [18]. In the current study, the colour value a* of the oat grains almost did not show any significant interactions with the chemical parameters of the heated oat flour, while high hue values were connected, to some extent, to the higher protein and ash content of non-heated and heated oat flours. The determination of the hue value of native grains would provide a simple and robust means to predict the protein and ash content of the oat batches.

## 5. Conclusions

Based on the current results, the properties of industrially produced oat flour could be predicted using the properties measured from the native grains prior to processing. The protein content determination of oat grains by NIT can be used to predict the chemical properties of oat flours after industrial-scale processing. In addition, the present data suggests that a high hectoliter weight and thousand seed weight are related to higher starch and lower protein contents of oat grains. The lightness of the native grains was connected to the chemical properties of the non-heated flours as well as the subsequent oat flour produced at the industrial scale. In the future, it would be interesting to study the relationship between the cultivar, growth conditions and the industrial milling behavior, to investigate the relationship of these findings to specific food applications, and to develop oat cultivars for specific food uses.

## Figures and Tables

**Figure 1 foods-10-01552-f001:**
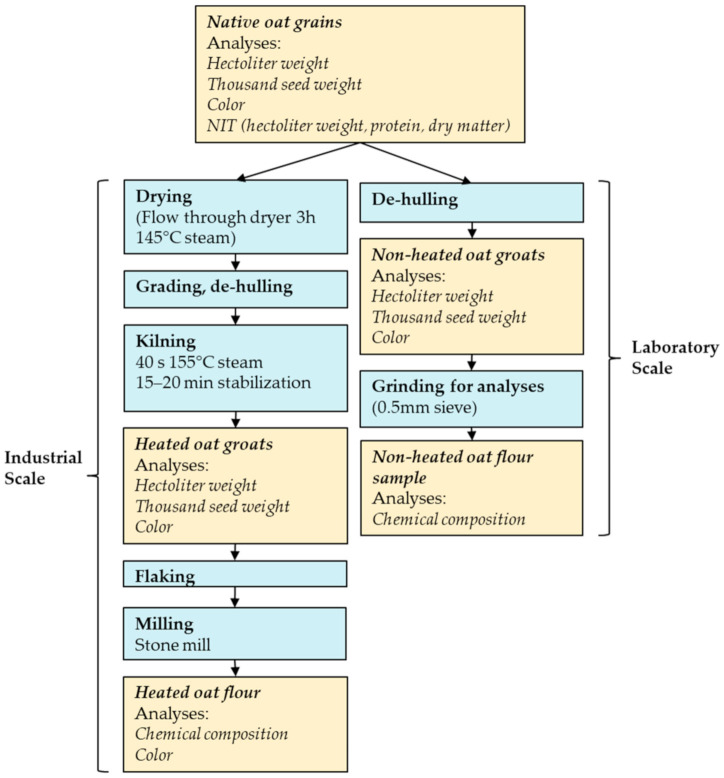
Processing scheme of the oat raw materials. The raw materials and subsequent analyses are shown in the yellow boxes, with the processing steps in blue.

**Figure 2 foods-10-01552-f002:**
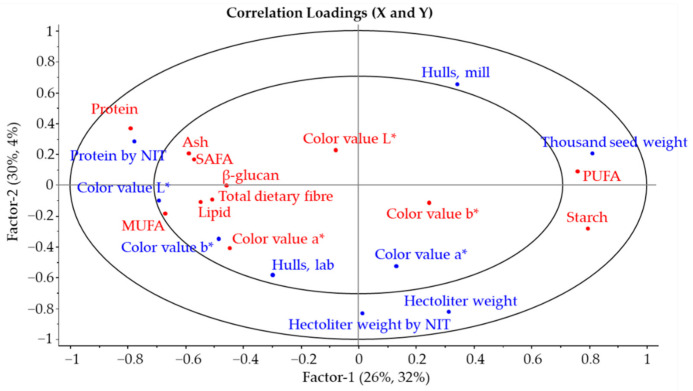
Partial least squares (PLS) regression projection of factors 1 and 2 of the native oat grain (blue) and heated oat flour (red) properties. The color values are L*, a* and b*; NIT, near-infrared transmittance; SAFA, saturated fatty acids; MUFA, monounsaturated fatty acids; PUFA, polyunsaturated fatty acids.

**Table 1 foods-10-01552-t001:** NIT (near infra-red transmittance) results of the oat grains, and the physical properties of the oat grains, non-heated groats and heated groats, presented as the average and range (min, max).

Parameter	Sample Type	Average	Range	Max
Hectoliter weight by NIT (kg/hL)	Grain	62.6 b	56.2	67.6
Hectoliter weight (kg/hL)	Grain	59.3 a	52.6	62.6
Hectoliter weight (kg/hL)	Non-heated groat	67.9 c	56.1	74.6
Hectoliter weight (kg/hL)	Heated groat	70.5 d	66.5	73.6
Thousand seed weight (g)	Grain	38.1 c	29.2	52.4
Thousand seed weight (g)	Non-heated groat	21.4 b	10.3	32.0
Thousand seed weight (g)	Heated groat	18.5 a	14.6	23.8
Hulls, lab (%)	Grain	44.2 x	14.1	67.3
Hulls, mill (%)	Grain	53.6 y	23.5	76.4

Different letters (a, b, c, d) within each column and parameter indicate statistically significant differences (*p* < 0.05) between the samples based on Tukey’s HSD test. Different letters, x and y, within each column and component indicate a statistically significant difference (*p* < 0.05) between the samples based on an independent sample *t*-test.

**Table 2 foods-10-01552-t002:** Color analysis results of 30 native oat grains, non-heat-treated groats, heat-treated groats and oat flour. The color values are the average values of 30 samples.

Sample Type	L*	a*	b*	Hue	Chroma
Grain	59.7 a	4.8 c	18.2 b	75.4 b	18.8 bc
Non-heated groat	61.6 b	5.0 d	17.8 b	74.4 a	18.5 b
Heated groat	64.9 c	4.4 b	18.7 c	76.7 c	19.2 c
Heated flour	89.2 d	0.8 a	8.2 a	84.5 d	8.2 a

Different letters (a, b, c, d) within each column and parameter indicate statistically significant differences (*p* < 0.05) between the Finnish samples based on Tukey’s HSD test. The grain color is the L*, a* or b* color spacer defined by the International Commission of Illumination (CIE).

**Table 3 foods-10-01552-t003:** Protein and dry matter content of 30 native oat grains, and the chemical composition of non-heated and heated oat flours as average and range values.

Component	Sample Type	Average	Range
NIT dry matter (%)	Grain	88.5	86.9–90.8
Dry matter (%)	Non-heated flour	89.7	87.0–92.3
Dry matter (%)	Heated oat flour	89.0	87.6–91.0
NIT Protein (dm, %)	Grain	12.6 a	9.1–15.8
Protein (dm, %)	Non-heated flour	15.6 b	11.4–20.5
Protein (dm, %)	Heated oat flour	15.3 b	10.6–19.2
Starch (dm, %)	Non-heated flour	60.6 x	46.6–75.3
Starch (dm, %)	Heated oat flour	63.4 y	57.8–71.9
Lipid (dm, %)	Non-heated flour	7.7 x	6.0–9.6
Lipid (dm, %)	Heated oat flour	7.3 x	5.4–9.4
SAFA (% of FA)	Non-heated flour	18.1 x	16.3–19.6
SAFA (% of FA)	Heated oat flour	18.1 x	16.1–19.7
MUFA (% of FA)	Non-heated flour	37.2 x	32.6–42.4
MUFA (% of FA)	Heated oat flour	37.1 x	33.0–42.2
PUFA (% of FA)	Non-heated flour	40.9 x	36.6–46.2
PUFA (% of FA)	Heated oat flour	40.4 x	36.5–43.9
Total dietary fibre (dm, %)	Heated oat flour	11.2	8.5–13.2
β-glucan (dm, %)	Non-heated flour	5.2 y	3.6–6.7
β-glucan (dm, %)	Heated oat flour	4.0 x	2.9–4.6
Ash (dm, %)	Non-heated flour	2.2 x	1.7–2.6
Ash (dm, %)	Heated oat flour	1.9 y	1.6–2.3

Different letters (a, b) within each column and parameter indicate statistically significant differences (*p* < 0.05) between the samples based on Tukey’s HSD test. Different letters, x and y, within each column and component indicate statistically significant differences (*p* < 0.05) between the samples based on independent sample *t*-tests. NIT, near infra-red transmittance; SAFA, saturated fatty acids; MUFA, monounsaturated fatty acids; PUFA, polyunsaturated fatty acids.

**Table 4 foods-10-01552-t004:** Pearson correlation coefficients of the oat grain and heated oat flour properties.

Flour	Grain	L*	Hue	Chroma	Protein by NIT	Hectoliter Weight	Thousand Seed Weight	Hulls, Lab	Hulls, Mill
L*		0.51 **	-	0.44 *	-	-	-	-	-
a*		-	-	-	0.37 *	-	−0.56 **	0.41 *	−0.49 **
Hue		-	-	-	−0.43 *	-	0.48 **	−0.45 *	0.44 *
Starch		−0.44 *	−0.46 *	-	−0.84 **	0.42 *	0.57 **	-	-
Protein		0.37 *	0.47 **	-	0.96 **	−0.41 *	−0.46 *	-	-
Lipid		0.37 *	-	-	0.41 *	-	−0.63 **	-	-
SAFA		-	-	-	0.51 **	−0.38 *	−0.44 *	-	-
MUFA		0.55 **	-	-	0.44 *	-	−0.64 **	-	-
PUFA		−0.54 **	-	-	−0.53 **	-	0.67 **	-	37 *
Ash		0.40 *	0.52 **	-	0.59 **	-	−0.38 *	-	-
Total dietary fibre		-	-	-	0.49 **	-	−0.38 *	-	-
β-glucan		-	-	-	0.50 **	-	−0.38 *	-	-

** Correlation is significant at the 0.01 level (two-tailed). * Correlation is significant at the 0.05 level (two-tailed). Color values: L*, lightness; a*, red–green scale. NIT, near-infrared transmittance; lab, laboratory scale; mill, industrial milling scale; SAFA, saturated fatty acids; MUFA, monounsaturated fatty acids; PUFA, polyunsaturated fatty acids.

## Data Availability

The data presented in this study are available on request from the corresponding author. The data are not publicly available due to agreement of the project.

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
