# Peer review of "Predicting the Properties of Industrially Produced Oat Flours by the Characteristics of Native Oat Grains or Non-Heat-Treated Groats"

_foods, 2021, doi:10.3390/foods10071552_

Round 1

Reviewer 1 Report

Abstract does not reflect the content of the article and needs to be corrected.

The authors of this paper did a great deal of research on a total of 47 oat grain samples and subsequent groat samples made from them. However, in my opinion, they were not able to handle the amount of research results obtained, their clear presentation and discussion.

Already at the beginning I have some concerns about the experimental design - 30 samples from Finland and 17 oat samples from other countries were anayzed, but by the different origins and varieties the authors found differences in chemical composition and physical properties due to the difference in the origin of the crop, which is obvious and should be expected. In addition, the issue was complicated by making by authors two types of groats from the grain samples and they were only subjected to certain determinations according to Tables 1 and 2. While the title of the article refers to the quality factors of the oat flour produced.

The explanation of what samples were tested and how the groats, flours were prepared is vague and chaotic. A diagram or table of the samples and a diagram of how the groats, flours were made with parameters are missing.

It is not stated in how many replicates the physicochemical and chemical tests were performed except for the colour and fatty acid tests. The authors state that they carried out TSW tests for grain (N) and unheated groats (NG) but in the table they give results for all types of groats. The results in the tables should be split, e.g. a separate table should be created for colour and a separate one for fatty acids. Missing, among other things, is the determination of the grain (N) without the use of NIT method as was done for groats.

The authors focused more on the characterisation of the oat samples collected and the groats produced. They charge the reader with the large number of correlations made, which, despite significance, are mostly weak or moderate. There is a lack of emphasis and exposure of the main aim of the study declared at the beginning and present in the title of the article.

The discussion also needs extensive revision. The authors describe that chemical and physical analyses showing results consistent with the literature. This shows that the results do not add anything new to the state of knowledge. There is a lack of discussion of the results in the context of the influence of quality of oat flour based on grain composition as suggested by the title of the article. Only in section 4.2. new information is given.

Figure 1 was made for oat grain and flour which would have supported greatly the discussion of the research results well but a small paragraph was devoted to it without discussing it.

Conclusions are missing. Instead of them, there is a repetition of information about what research has been done and what research is being done and recommended for the future.

I suggest considering to select research results from so many obtained, and focus on them in the results presentation and discussion. The idea of the article needs to be refined.

Author Response

Dear reviewer,

We are thankful for your insightful and comprehensive comments regarding our manuscript and we have now revised it accordingly. The aim of the current research was to study the relationships between the native oat grains, non-heat-treated oat groats and industrially produced oat flour.  It is evident that we have failed to clarify this aim in the original manuscript. Oats used in food industry usually are processed with the milling process unique for the oats, and therefore our target was to understand how the parameters measured from whole grains and groats correlate with the results with the flour properties.

We have made major revisions in all the sections of the manuscript, as specified in the following paragraphs:

The abstract has been modified to describe more accurately the contents of the manuscript.

We agree that the research frame was not described clearly enough. To tackle this issue, we have added a processing schematic of the samples (new Figure 1). We hope that this will clarify how the samples were produced.

Thank you for your observations regarding the number of replicates as well as the presentation of the results. We have now added more detailed description of the replicates to the materials and methods section. The tables in the results section have been modified in order to make them more easily readable. We did not determine the chemical composition of the native grains as it is not considered relevant to this study. Quality properties, such as hectoliter weight and color, are usually measured upon delivery to mills from the whole hulled grains. In addition, oats are not suitable for food use without de-hulling; besides there is already existing data of the chemical composition oat grains before de-hulling and milling process.

Your point regarding the presentation of the correlations in the result section is valid. In order to provide more accurate description of the correlations, we have removed some of the weaker correlation descriptions from the text and described the relevance of the correlations.

The discussion section has been revised and reorganized extensively to respond more specifically to the study aim and approach. For instance, as you kindly noted, highlighting the earlier literature is not necessary to this extend and have removed some of the unnecessary sentences. Furthermore, we have included the discussion about the figure 1 (currently figure 2) as it supports the results of our research well as you noted. Conclusions section has been modified to provide the clear and compact description of the main findings.

After careful revision, we decided to remove the data regarding the oat samples from UK, USA and Canada, as they were not processed in the industrial scale and thus the data does not support the main aim. By doing this, we believe we have accomplished a well-readable manuscript with novel information on oats.

Sincerely, on behalf of the co-authors,

Iina Jokinen

Reviewer 2 Report

The present manuscript named ‘Quality factors of industrially produced oat flours in relation to the composition of the native grains’ is not-well written although it contains useful scientific data that will be useful to the industry. The study was not well-structured. There are a lot of results that are not presented correctly, so the conclusions are not justified the aim of the present study.

The description of materials and methods is confused and in some cases is missing. There are a lot of abbreviations so it is difficult to read all the text without going to the previous section. I feel that the English language should be improved. All the Tables are confused and they should modified correctly in order to present clearly the data. Additionally, many samples were not analysed by chemical analyses (Table 2).

Other comments:

Lines 56-58. Refresh all the sentence.

Line 104. What means N and NG?

Line 110. The calibration was done only with white ceramic plate?

Line 129. ‘fat content20011??????

Lines 130-132. I don’t think it is necessary to mention it.

Line 145. What is 68D?

Line 146. Expalin what is FAs

Table 1. All the data is very confused. There are a lot of abbreviation and it is difficult to read the results. Refresh the title….there are many and…and..and……Put the abbreviations at the bottom of Table.

Author Response

Dear reviewer,

Thank you for your accurate and concise comments regarding our manuscript “Quality factors of industrially produced oat flours in relation to the composition of the native grains”. We appreciate that you have pointed out the significance of the data for the industry. Furthermore, we agree that the original manuscript failed to provide a clear representation of the aims and results of the current study. The aim of the current research was to study the relationships between the native oat grains, non-heat-treated oat groats and industrially produced oat flour. To correct this, we have carefully revised and modified all the sections of the manuscript.

Shortly, the study aim was modified and re-structured in order to help the reader to follow the key idea of the research. Further, some modifications were made to the introduction in order to support the aim. To clarify the methodology, a processing scheme was added to show the sampling and related analyses for the different grain, groat and flour samples. Most of the abbreviations have been removed and replaced with whole words. The result section was modified by removing the unnecessary data, by simplifying the tables and by simplifying the description of the correlation results i.e. removing some of the weaker correlations from the text as they are presented in the table. The discussion section has been revised and reorganized extensively to respond to the aims of the study. Similarly, the title has been revised to describe more accurately the contents of the manuscript.

Answers to the specific comments:

Lines 56-58. Refresh all the sentence.

The sentence has been revised and restructured in lines 67-69 (track-changes on).

Line 104. What means N and NG?

The N and NG abbreviations representing oat grains and oat groats have been removed and replaced with whole words in line 137 (track-changes on).

Line 110. The calibration was done only with white ceramic plate?

The calibration of the Minolta instrument was done with ceramic plate as it is the standardized calibration method used for the instrument for all sample types.

Line 129. ‘fat content20011??????

Line xx. ‘fat content.2011’ has been corrected to “fat content.1999” in line 166 (track-changes on)..

Lines 130-132. I don’t think it is necessary to mention it.

This sentence refers to the accreditation of the proximate composition methods. Indication of accredited methods means that the results were acquired within laboratory´s quality assurance and quality control procedures. Therefore, we see it is important to keep it in the text.

Line 145. What is 68D?

68D is the name of the fatty acid methyl ester standard. We have added “FAME standard” before the name 68D to clarify this issue in line 185(track-changes on).

Line 146. Expalin what is FAs

Lines 184-185 (track-changes on). We decided to remove this sentence as it was not seen relevant.

Table 1. All the data is very confused. There are a lot of abbreviation and it is difficult to read the results. Refresh the title….there are many and…and..and……Put the abbreviations at the bottom of Table.

Table 1. Most of the abbreviations have been removed and remaining ones have been transferred to the bottom of the table. Additionally, the table has been split into two (Tables 1 and 2) in order the provide clearer presentation of the results.

Sincerely, on behalf of the co-authors,

Iina Jokinen

Reviewer 3 Report

Dear Authors,

I revised the paper “Quality factors of industrially produced oat flours in relation to the composition of the native grains” with pleasure.

The paper is well written. Introduction is well constructed, methods are sound, results are satisfactory and conclusions are supported by results.

Few amendments are required as below:

Please, include NIR in keywords;

Line 80: based on which criteria were samples “chosen”?

Line 93: please, replace “0,5 mm” with “0.5 mm”

Line 104: please, define “N” and “NG” in the text.

Paragraph 2.1: it is not clear how many types of samples were studied. Were Finnish samples analysed as native grains, non-heat-treated groats and heat-treated groats/oat flours? Were international samples analysed as native oat grains and non-heat-treated oat groats? I suggest adding a scheme reporting sampling and the type of analyzed samples.

Line 212-213: The sentence is not clear. Please, check and amend.

Table 1 and Table 2: please, add standard deviation values

Line 392: You stated that differences between Finnish and International samples were expected since geographical location affects oat composition. Do you expect that international samples from other countries might have a composition similar to Finnish samples? Why did you choose samples from Canada, USA and UK?

Author Response

Dear reviewer,

Thank you for reviewing our manuscript “Quality factors of industrially produced oat flours in relation to the composition of the native grains”.

We have reviewed the amendments you have provided and revised the manuscript accordingly.

Replies for the presented amendments:

Please, include NIR in keywords;

We believe the reviewer means NIT (nor NIR). NIT has been included into the key words.

Line 80: based on which criteria were samples “chosen”?

Samples were chosen based on the availability during that season as well as the expectation that the samples have differing properties. This description has been added into the manuscript and can be found in lines 101-102 (track-changes on).

Line 93: please, replace “0,5 mm” with “0.5 mm”

This has been corrected on line 118 (track-changes on).

Line 104: please, define “N” and “NG” in the text.

After revising the manuscript, we decided to remove all the unnecessary abbreviations and leave only the well-established ones. Therefore, these two abbreviations were opened as “grains” and “non-heat-treated groats” in line 137 (track-changes on). 

Paragraph 2.1: it is not clear how many types of samples were studied. Were Finnish samples analyzed as native grains, non-heat-treated groats and heat-treated groats/oat flours? Were international samples analysed as native oat grains and non-heat-treated oat groats? I suggest adding a scheme reporting sampling and the type of analyzed samples.

We agree that the research frame was not described clearly enough. To tackle this issue, we have added a processing schematic of the samples. We hope that this will clarify how the samples were produced. International samples were anlaysed only as native grains and non-heat-treated groats and after careful revision, we decided to remove the data regarding the international samples as they were not processed in the industrial scale and thus the data does not support the main aim.

Line 212-213: The sentence is not clear. Please, check and amend.

This has been now revised and clarified on lines 268-269 (track-changes on).

Table 1 and Table 2: please, add standard deviation values

Since we calculated the statistical significance between the values as indicated with letters in the Tables 1, 2 and 3 and to avoid complicating the tables, we did not add the standard deviation values here. However, the standard deviation are shown in the supplementary materials (Appendix A, Tables A1, A2, A3 and A4).

Line 392: You stated that differences between Finnish and International samples were expected since geographical location affects oat composition. Do you expect that international samples from other countries might have a composition similar to Finnish samples? Why did you choose samples from Canada, USA and UK?

After careful revision, we decided to remove the data regarding the oat samples from UK, USA and Canada, as they were not processed in the industrial scale and thus the data does not support the main aim. These samples were chosen in order to have some comparison with oats from other geographical locations. 

Sincerely, on behalf of the co-authors,

Iina Jokinen

Round 2

Reviewer 1 Report

The authors have done a very good job. They have made many profound corrections and thus obtained a completely new article. After the corrections made and the excess of results removed, the article has become clearer in reading the content.

The idea of changing the title of the paper was a very good move. The new title of the paper better reflects the substantive content of the article. Also the abstract has been better formulated. The inclusion of a diagram of the processing of the samples with the marked determinations has made a big difference.

The authors provided the missing number of repetitions for the performed physicochemical and chemical tests.

The discussion of the results of the article now clearly refers to the purpose stated in the title. It has been significantly improved and indicates the originality of the work.

The conclusions have also been clearly improved and are given based on the results and discussion of the results. This section no longer contains an abstract of the paper.

The authors have written the paper that is much better than the original version, and I see no significant flaws in the article, so I recommend that the article be accepted for publication in the journal in its current form.

Author Response

Dear reviewer,

Thank you for reviewing our revised manuscript. We are pleased to hear that the corrections made after the first review round have provided satisfactory changes to the manuscript and fully agree that the manuscript was significantly improved after these corrections.

Thank you for the review process.

Sincerely, on behalf of the co-authors,

Iina Jokinen

Reviewer 2 Report

The authors have improved the quality of manuscript in the revised version.

I don't have any other commnets.

Author Response

Dear reviewer,

Thank you for reviewing our revised manuscript. We are pleased to hear that the correction made after the first review round have provided satisfactory changes to the manuscript.

Thank you for the review process.

Sincerely, on behalf of the co-authors,

Iina Jokinen